# Quantitative Structure–Neurotoxicity Assessment and In Vitro Evaluation of Neuroprotective and MAO-B Inhibitory Activities of Series *N*′-substituted 3-(1,3,7-trimethyl-xanthin-8-ylthio)propanehydrazides

**DOI:** 10.3390/molecules27165321

**Published:** 2022-08-20

**Authors:** Magdalena Kondeva-Burdina, Javor Mitkov, Iva Valkova, Lily Peikova, Maya Georgieva, Alexander Zlatkov

**Affiliations:** 1Laboratory of Drug Metabolism and Drug Toxicity, Department of Pharmacology, Pharmacotherapy and Toxicology, Faculty of Pharmacy, Medical University of Sofia, 2 Dunav Street, 1000 Sofia, Bulgaria; 2Department of Pharmaceutical Chemistry, Faculty of Pharmacy, Medical University of Sofia, 2 Dunav Street, 1000 Sofia, Bulgaria; 3Department of Chemistry, Faculty of Pharmacy, Medical University of Sofia, 2 Dunav Street, 1000 Sofia, Bulgaria

**Keywords:** xanthine, monoamine oxidase B, neurotoxicity, QSTR, neuroprotection

## Abstract

The neurotoxic, neuroprotective and MAO-B inhibitory effects of series *N*′-substituted 3-(1,3,7-trimethyl-xanthin-8-ylthio)propanehydrazides are evaluated. The results indicate compounds *N*′-(2,3-dimethoxybenzylidene)-3-(1,3,7-trimethyl-2,6-dioxo-2,3,6,7-tetrahydro-1*H*-purin-8-ylthio)propanehydrazide (**6k**) and *N*′-(2-hydroxybenzylidene)-3-(1,3,7-trimethyl-2,6-dioxo-2,3,6,7-tetrahydro-1*H*-purin-8-ylthio)propanehydrazide (**6l**) as most perspective. The performed QSTR analysis identified that the decreased lipophilicity and smaller dipole moments of the molecules are the structural features ensuring lower neurotoxicity. The obtained results may be used as initial information in the further design of (xanthinyl-8-ylthio)propanhydrazides with potential *h*MAOB inhibitory effect and pronounced neuroprotection.

## 1. Introduction

Parkinson’s disease (PD) is a neurodegenerative disease that mainly affects dopamine-producing (“dopaminergic”) neurons in a specific area of the brain called the substancia nigra and results in a reduction in dopamine plasma levels. It is a long-term progressive degenerative syndrome that mainly affects the motor system, and its conditions are treated mainly through the amendment of the symptoms with l-DOPA and/or DA agonists [1].

Monoamine oxidase B (MAO-B) (an enzyme containing flavin adenine dinucleotide (FAD)) catalyzes the oxidative deamination of dopamine—one of its main catabolic reactions in the brain. MAO-B inhibitors, such as Selegiline (l-Deprenyl), are useful in the treatment of Parkinson’s disease (PD) because they may preserve the depleted dopamine supply and elevate the levels of dopamine produced for exogenously administered levodopa [2]. Selegiline is an irreversible but not highly selective MAO-B inhibitor administered to gain the l-DOPA level in Parkinson therapy as well as to reach a protective effect in patients with the pre-Parkinson syndrome [3] due to reducing the concentrations of potentially hazardous by-products, such as dopaldehyde and H_2_O_2_ produced by MAO-B-catalyzed dopamine oxidation [4]. It also was found that small molecules consisting of two electron-rich moieties connected via a short spacer were shown to be potent MAO A/B inhibitors [5].

Caffeine is an adenosine analogue that acts as a non-selective adenosine receptor antagonist [6] and exhibits a weak MAO-B inhibitor activity (Ki = 3.6 mM) [7]. The majority of the effects of caffeine are mainly mediated by the blockade of adenosine receptors, and the proved neuroprotective effects of caffeine in brain disorders have been mimicked by the blockade of adenosine A_2A_ receptors (A_2A_Rs) [8,9].The adenosine A_2A_ receptor has been identified as an important and attractive target for nondopaminergic therapy in PD [10,11,12,13,14]. Some recent studies have shown that A_2A_ receptor blockades, especially by specific adenosine receptor antagonists, may have a statistically significant neuroprotective effect on dopaminergic neurodegeneration [15,16,17]. The inhibition strength of caffeine is considerably increased by substitution at C-8 of the caffeinyl ring [7,18,19]. The structures of some potent reversible inhibitors of human monoamine oxidase (MAO) B are shown in Figure 1.

Hydrazide-hydrazones are not only intermediates but they are also very effective organic compounds and have been proposed as promising scaffolds with a large number of pharmacological actions, including an MAO inhibitory property, as new multi-target-directed ligands aimed to be effective against neurodegenerative diseases [20]. Within the last years, a small number of new compounds were synthesized and tested as MAO inhibitors keeping constant the common CONHN=CH linker and varying the two opposite tails: the former comprehended electron-rich moiety, whereas the hydrazonic nitrogen was functionalized with mono- or disubstituted aryl groups [21,22].

Based on the currently presented data on the effectiveness of these functional compartments, the design of such MAO-B inhibitors, containing both a xanthine cycle and hydrazide-hydrazone moiety in their structure, has become a strongly researched area to identify suitable drug candidates for the treatment of neurological diseases, particularly PD.

This determined the purpose of this study to synthesize a group of xanthine-based hydrazide-hydrazones and to evaluate their neurotoxicity and neuroprotective properties, along with an assessment of their possible MAO-B inhibitory effects through appropriate in vitro methodologies. In addition, an elucidation of the structural fragments of the molecules was obtained through an in silico QSTR methodology.

## 2. Results

### 2.1. Chemistry

A standard common synthetic pathway for synthesis of the investigated hydrazide-hydrazones was used [23]. The tested compounds contained different substituents (OH, OCH_3_, NO_2_, Cl) at the aldehyde fragment, and the number, type and position of the substituents (Table 1) were also varied.

All obtained compounds were characterized by instrumental analytical procedures (UV-Vis, FTIR, NMR, MS) and the results were found to correspond to the assigned structures [23].

The analysis of the structural features defined the dichloro-substituted derivative **6c**, along with the *m*,*p* di- (**6g**) and tri-methoxy (**6h**)-substituted analogues and *m*,*p* di- (**6i**) and tri-hydroxy-substituted methoxy derivative (**6j**) as the most polar structures.

### 2.2. Neurotoxicity Evaluation

The neurotoxicity studies of the target derivatives were performed on subcellular fractions isolated from rat brain homogenate-synaptosomes, microsomes and mitochondria.

In all neurotoxicity evaluations, caffeine, identified in all figures as compound **1**, was used as a model substance for comparison and neurotoxicity evaluation of the newly synthesized derivatives, based on caffeine core as an initial structural fragment.

#### 2.2.1. Effects of the Evaluated Hydrazide-Hydrazones on Isolated Rat Brain Synaptosomes

The performed in vitro toxicological screening was based on monitoring the change in cell viability and reduced glutathione levels in subcellular fraction. The evaluated rat synaptosomes were isolated from brain tissue through homogenization followed by size or density-based fractionation of the homogenate.

Figure 2 is presenting the results obtained after a single application of compounds **6**, **6a**–**o** in a 100 µmol concentration on the cellular viability of the isolated synaptosomes. From the evaluated series, only **6k** and **6l**, as well as the initial caffeine (**1**), showed the lowest neurotoxicity, when compared to the control (untreated synaptosomes). The rest of the structures showed more pronounced toxicity than caffeine.

The results demonstrated that **6k** decreases the synaptosomal viability by 29%, **6l** by 28% and caffeine by 26% against the control (untreated synaptosomes).

#### 2.2.2. Effects of the Evaluated Hydrazide-Hydrazones on Glutathione (GSH) Level in Isolated Rat Brain Synaptosomes

Another main indicator used for preliminary evaluation of subcellular toxicity is the influence of the tested compounds on the levels of reduced glutathione in the synaptosomal fraction.

On this parameter for the evaluated series, when applied alone at a 100 µmol concentration compound, **6k** decreased the GSH level by 30%, **6l** by 25% and **1** by 25% against the control (untreated synaptosomes) (Figure 3).

When administered alone, compounds **6**, **6a**–**o** were found to show higher toxicity than the initial reagent caffeine in isolated rat synaptosomes, affecting in the same way both evaluated parameters as demonstrated on Figure 2 and Figure 3.

#### 2.2.3. Effects of the Evaluated Hydrazide-Hydrazones on Isolated Rat Brain Microsomes

When tested alone in a concentration of 100 µmol, from the evaluated series **6**, **6a**–**o** on isolated rat brain microsomes, only **6k** and **6l**, as well as caffeine, showed the lowest neurotoxicity, when compared to the control (untreated microsomes). The rest of the structures performed more pronounced toxicity than caffeine (Figure 4).

It was determined that **6k****, 6l** and caffeine did not show any statistically significant pro-oxidant effect compared to the control (untreated microsomes) (Figure 4).

#### 2.2.4. Effects of the Evaluated Hydrazide-Hydrazones on Isolated Rat Brain Mitochondria

The least toxic representatives from the newly synthesized xanthine derivatives **6**, **6a**–**o** were subjected to an in vitro evaluation of their effects on isolated rat brain mitochondria.

The results from the evaluation of the effects of the tested *N*′-substituted 3-(1,3,7-trimethyl-xanthin-8-ylthio)propanehydrazide-hydrazones on the malonaldehyde (MDA) production and the GSH level in isolated rat brain mitochondria are presented on Figure 5 and Figure 6, respectively.

When applied alone, in a concentration of 100 µmol, **6k****, 6l** and caffeine did not change statistically significantly the MDA production, against the control (untreated mitochondria) (Figure 5).

On the other evaluated parameter (GSH level), only **6l** did not show any statistically significant neurotoxic effect. Compounds **6k** and 1 decreased the GSH production by 25% and by 20%, respectively, against the control (untreated mitochondria).

On brain mitochondria alone, **6k**, **6l** and caffeine did not change statistically significantly the production of malondialdehyde (MDA) compared to controls (untreated mitochondria). However, at the GSH level, only **6l** did not show a statistically significant neurotoxic effect.

### 2.3. Quantitative Structure–Neurotoxicity Assessment for the Evaluated N′-Substituted Xanthinylthio Propanehydrazones

A quantitative structure–neurotoxicity relationship analysis (QSTR) was performed in an attempt to identify structural features of the newly synthesized hydrazones **6a**–**p** that affect their neurotoxicity. Relative synaptosomal viability (RV) and relative increase in MDA production (RMDA), calculated as described in Section 4.3., were used as quantitative characteristics of the neurotoxicity of compounds. Caffeine (**1**) was chosen as a reference compound. It possessed the lowest neurotoxicity and assumed a value of one for both RV and RMDA. All other compounds were more toxic, so they acquired lower values of RV and higher RMDA values, respectively. RV and RMDA, approaching the unit reference value of caffeine, were considered as indicators for improvement in the neurotoxicity profile, as is observed in **6k** and **6l**.

Descriptors reflecting the electronic (HOMO, LUMO, atomic charges of common heteroatoms) and molecular properties (molecular weight, number of rings, number of H-bond donors and acceptors, etc.) of compounds as well as 3D descriptors (volume, surface, ovality, dipole, etc.) were calculated for the purposes of study. A genetic algorithm (GA) procedure was applied for selection of relevant descriptors. They were further used in the development of multiple regression (MLR) models for the assessment of the relationship between the structure and neurotoxicity of compounds. The values of RV, RMDA and the descriptors included in the final equations are presented in Table 2.

#### 2.3.1. QSTR Model Based on Data for the Relative Synaptosomal Vitality (RV)

The GA regression generated the following equation for viability of brain synaptosomes:RV = −0.05143 × *logP_ACD* − 0.005916 × *Dipole* + 1.0033      *n* = *15 r*^2^ = *0.608* *SEE* = *0.063 F* = *9.32 q*^2^ = *0.523 r*^2^*_scr_* = *0.144*(1)

The descriptor *logP_ACD* accounts for lipophilicity of compounds. Its value increases when hydrophobic substituents are introduced in the structure and decreases in presence of hydrophilic ones. The negative regression coefficient indicates that higher *logP_ACD* values correspond to lower RV, i.e., to a higher neurotoxicity of compounds.

The dipole moment of molecules *Dipole* (Debye) is calculated as a vector sum of the individual moments of bonds. It depends on bond polarity, i.e., on electronegativity of bound atoms, as well as on the geometry of molecules and their symmetry. *Dipole* values, for the studied compounds, were in the range between 2657 D and 21,067 D (Table 2). The parameter was negatively correlated to RV, which means the molecules of higher polarity reduce more significantly the cell viability.

The derived QSTR model was considered as statistically significant on the basis of explained variance *r*^2^, standard error of estimate *SEE* and *F*-ratio. The results from the Y-scrambling test (*r*^2^*_scr_* = *0.144*) indicated a very low fitting performance of the permuted model and suggested a reduced risk of chance correlations. The leave-one-out cross-validation procedure demonstrated the moderate but still acceptable predictive ability of the model (*q*^2^ = *0.523*). Correlation between the experimentally determined and calculated-by-the-model RV values is presented on Figure 7. Only one compound (**6l**) was identified as an outlier.

#### 2.3.2. QSTR Model Based on Data for the Relative Increase in Microsomal MDA Production (RMDA)

After selection of the descriptors by GA and linear regression analysis, the following two parameter equation was obtained:RMDA = 0.8115 × *xc3* + 0.03691 × *Dipole* − 0.25712          *n* = *15 r*^2^ = *0.600 SEE* = *0.382 F* = *9.751 q*^2^ = *0.4553 r*^2^*_scr_* = *0.153*(2)


The parameter *xc*3 is a topological descriptor from the group of molecular connectivity *χ* indices. It is a simple *χ* cluster index of a third-order and reflects the presence of branched fragments, consisted of three heavy atoms connected to a central atom. *xc*3 values are calculated by summing all the subgraphs of this order and type that are found in the molecular graph. Values of this parameter for the tested compounds varied between 1.253 and 3.379. It increases when the number of substituents in the benzene nucleus increases as well as when substitutions at *p*- or *m*, *p*-positions are made. Mono- and disubstituted derivatives at *m*- and *o*, *m*-positions possess lower *xc*3 indices. Data for microsomal MDA production follow the same trend. The meaning of the parameter *Dipole* was already discussed in the previous section.

Both descriptors in the equation have positive regression coefficients. Higher values of *xc3* and *Dipole* increase RMDA and are associated with an increased neurotoxicity.

The derived QSTR model did not meet that discussed in the QSAR society criteria for the statistical significance and predictivity of models [24]. Although it could not be used for RMDA prediction of non-synthesized compounds, it still allowed some qualitative interpretations of results.

The correlation between experimentally determined and calculated values of RMDA is presented on Figure 8.

### 2.4. Evaluation of the Neuroprotection Effects of the Studied N′-substituted 3-(1,3,7-trimethyl-xanthin-8-ylthio)propanehydrazides

#### 2.4.1. Protective Effects in a Model of 6-OHDA-Induced Oxidative Stress in Isolated Rat Synaptosomes

The least toxic derivatives from the tested series were subjected to an evaluation of the possible neuroprotective effects in a model of 6-OHDA-induced oxidative stress, determining the influence on two parameters—synaptosomal vitality and GSH level.

The results from the measurements of the synaptosomal viability presented on Figure 9 indicate that only compounds **6k** and **6l** showed a statistically significant protective effect in the model of 6-OHDA-induced oxidative stress, where **6k** protected the synaptosomal vitality by 20%, and **6l** by 36%, against the toxic agent. Caffeine stored the synaptosomal vitality by 109% against the toxic agent.

From the analyses performed on the neuroprotective effects of the lowest toxic compounds in increasing the cell viability of isolated rat synaptosomes, compound **6l** was characterized by the best protective effects in 6-OHDA-induced oxidative stress.

When analyzing the influence of the target compounds **6k** and **6l** on the reduced glutathione levels in the model of 6-OHDA-induced oxidative stress, it was determined that **6k** and **6l** showed statistically significant protective effect, with **6k** storing the GSH levels by 20%, and **6l** by 42%, against the toxic agent (Figure 9).

Caffeine stored the GSH levels by 183% against the toxic agent (Figure 9). Pure caffeine protected the GSH levels by 30%, against 6-OHDA.

When applied alone, 6-OHDA performed a pronounced neurotoxic effect by decreasing the GSH levels by 50%, against the control (untreated synaptosomes). On this parameter, compound **6l** performed the most pronounced protective effect, comparable to the one of caffeine.

The results obtained for the effects of the tested compounds on the second evaluated parameter—GSH level, characterizing the functional-metabolic status of isolated rat synaptosomes—largely confirmed the results obtained by analyzing the effect of substances on synaptosomal viability, with **6k** and **6l** performing a statistically significant protective effect in 6-OHDA-induced oxidative stress on both parameters.

#### 2.4.2. Protective Effects in a Model of Iron Ascorbate (Fe^2+^/AA)-Induced Lipid Peroxidation in Isolated Rat Microsomes

On its own, iron ascorbate (Fe^2+^/AA) has a pronounced pro-oxidant effect, increasing MDA production by 273% compared to controls (untreated microsomes) (Figure 10).

In this model, **6k** reduced MDA production by 38% and **6l** by 50%, compared to the toxic agent. Pure caffeine reduced MDA by 39%, compared to Fe^2+^/AA (Figure 10).

#### 2.4.3. Protective Effects in a Model of Tert-Butylhydroxyperoxide(t-BuOOH)-Induced Oxidative Stress in Isolated Rat Mitochondria

For this evaluation, the MDA production and GSH levels were used as quantitative parameters to measure the effect in isolated rat mitochondria. The results obtained are presented on Figure 11.

Again, **6l** exhibited a more pronounced effect than the one of **6k** and caffeine itself (Figure 11), where self-administered *t*-BuOOH showed a pronounced neurotoxic effect by increasing MDA production by 55% and decreasing GSH by 52%, compared to controls (untreated mitochondria).

In this neurotoxicity model, **6k** decreased the MDA production by 50% and maintained the GSH level by 17%, while **6l** recorded 61% and 54%, respectively, against the toxic agent. Caffeine decreased MDA and preserved GSH level by 56% and by 40%, respectively, against pure *t*-BuOOH.

### 2.5. Investigation of the Influence of the Newly Obtained Derivatives on the Activity of Human Recombinant MAO-B Enzyme

Thus, it was of significant interest to establish the possible MAO B inhibitory effects of the newly synthesized xanthine derivatives.

Only substances **6k**, **6l** and **1** showed a statistically significant inhibitory effect on the activity of the *h*MAOB enzyme, when applied alone, which is close to that of Selegiline, a classic MAO-B inhibitor. For the most active structures, it was found that **6k** inhibited the enzyme by 25%, **6l** by 30% and caffeine by 26%, compared to Selegiline, inhibiting *h*MAOB by 55% (Figure 12).

As the molecule with the best inhibitory activity from the tested series, the results outlined the new *N*′-(2-hydroxybenzylidene)-3-(1,3,7-trimethyl-2,6-dioxo-2,3,6,7-tetrahydro-1*H*-purin-8-ylthio)propanehydrazide (**6l**).

## 3. Discussion

### 3.1. Neurotoxicity Evaluation

The brain is particularly sensitive to toxicity due to its high metabolic activity and its limited ability to regenerate. Potential neurotoxic effects of xenobiotics can be expected due to increased neurotransmitter levels, enhanced functional activity at the respective receptors or direct cellular toxicity, leading to mitochondrial dysfunction, apoptosis or inhibition of neurogenesis [25].

The isolated rat synaptosomes have a large number of released postsynaptic formations mixed with the presynaptic part of the synapses and are an important source of information on neurotransmitters, cell membrane polarization and ion exchange. This allows their use as an in vitro system for assessing the degree and rate of release of neurotransmitters in the CNS [26]. The other main indicator of preliminary information on the presence of subcellular toxicity is the determination of the level of reduced glutathione in the synaptosomal fraction [27].

It is noteworthy to mention that based on the results from the evaluation of the effects on the GSH levels, only substances **6k**, **6l** and caffeine had the lowest neurotoxicity compared to the control (untreated synaptosomes).

Microsomes proved to be the most suitable model for studying drug metabolism and its effect on the brain. They are small, sealed vesicles that originate from fragmented cell membranes (often the endoplasmic reticulum) and contain specific individual human metabolizing enzymes, such as cytochrome (CYP) and uridine-5’-diphosphoglucuronosyl transferase (UGT). This makes them a suitable model for in vitro studies of drug metabolism, in particular for the study of the contribution of enzymes involved in the biotransformation of drugs and xenobiotics [28].

In view of the above, a probable toxic effect on isolated rat brain microsomes was also investigated for the test compounds. The effect of newly synthesized molecules on malonaldehyde (MDA) production in isolated brain microsomes was used as a quantitative parameter for neurotoxicity, which identified compounds **6k**, **6l** and caffeine, to express the lowest toxicity in this subcellular fraction.

The analysis of isolated brain synaptosomes and microsomes showed that most of the newly synthesized caffeine derivatives and caffeine showed a statistically significant neurotoxic effect compared to the control (untreated synaptosomes and microsomes), where only compounds **6k** and **6l** showed a lower statistically significant neurotoxic effect.

Mitochondria are unique organelles with major role in cellular function with a central role in determining the point-of-no-return for the apoptotic process. Literary data identify the advantage of isolated mitochondria based on the lack of complexity of the freshly isolated cells possessing the barrier of membrane permeability and an active metabolizing system. Currently, the analysis of mitochondrial function has become central to the basic research of mitochondrial physiology and the diagnosis of many diseases including cancer, diabetes, cardiovascular disease, oxidative stress and the age-related neurodegenerative diseases [29].

The results from the evaluations on this parameter confirmed the neurotoxicity properties identified for the representatives in the tested series and extracted compounds **6k** and **6l** as least toxic at a subcellular level. The data suggest that the introduction of a –OCH_3_ and/or –OH group at *o*-position relative to the hydrazone group in the phenyl core of the carbonyl fragment in the structure is optimal for a decrease in the toxicity manifested at a subcellular level. An appearance of an additional –OCH_3_ group at *m*-position or replacement of the *o*-substituent with more electronegative Cl atom leads to a formal increase in the subcellular toxicity performed by the target xanthinylthio-hydrazide hydrazone derivatives.

### 3.2. Quantitative Structure–Neurotoxicity Assessment for the Evaluated Xanthinylthio Propanehydrazones

The present QSTR study was directed toward the development of models suitable to analyze and understand the relationship between the structure and neurotoxicity of the newly synthesized hydrazones **6a**–**p.** Even designed as a preliminary assessment of the key properties and structural characteristics of compounds related to their neurotoxicity, it would be helpful in the prospective for our future investigations.

The model represents a two parameters equation, reflecting the relationship between the lipophylicity and polarity of compounds, and their capability to suppress the vitality of brain synaptosomes. The more lipophilic compounds, **6c**, **6d**, **6e**, **6f** and **6m** (*logP_ACD* > 4), being Cl, Br, CH_3_ and CF_3_ derivatives, more strongly decrease the cell viability. The least toxic, **6k** and **6l,** are hydroxyl- and methoxy-analogues (*logP_ACD* ≤ 3.5). The most polar derivatives, **6c** (dichloro-substituted), **6g** (*m*, *p*-dimethoxy), **6h** (trimethoxy), **6i** (*p*-hydroxy, *m*-dimethoxy) and **6j** (*p*-hydroxy, *m*-methoxy), appeared to be amongst the most toxic compounds in the series. Respectively, the least toxic **6l**, as well as caffeine, had the lowest *Dipole* values.

The derived QSTR model was estimated as statistically significant because it explained about 61% of variability in the target variable RV [24]. Despite the external validation with an independent test set being the most reliable procedure for quality assurance of the model [30], in our case Y-scrambling was chosen, because of the small number of compounds. Randomization of RV confirmed the model’s usefulness in understanding and explaining the structure–RV relationship. The model performed satisfactorily in predicting the RV of the tested compounds and only one outlier, **6l,** was identified. It was the only monosubstituted hydroxyl derivative in the set, which although the least polar, has an intermediate lipophilicity.

Finally, the finding that the hydrophobic and strongly electronegative substituents increase the neurotoxicity of compounds would be implemented in our further investigations. As far as **6l** being the least toxic compound in the series, it would be considered as a suitable parent compound in a future synthesis.

The derived model did not meet the criteria for the statistical significance and predictivity of models, namely *r*^2^ > *0.6* and *q*^2^ > *0.5* [24], which allowed only some qualitative remarks. The model outlined again the role of the polarity of molecules in the neurotoxicity, manifested by studied compounds. It also suggested the optimal number (up to two) and positions (preferably *m*- or *o*, *m*-) of substituents in the phenyl ring. That information, although qualitative, corresponds to the findings, discussed in the previous section.

### 3.3. Studies on the Neuroprotective Activity of Newly Synthesized Derivatives

Neuroprotection is a process of relative preservation of neural structure and/or function [31]. It aims to prevent or slow the progression of disease or subsequent damage by preventing or slowing down the loss of neurons [32]. Despite the differences in symptoms and impairment associated with CNS disorders, many of the mechanisms of neurodegeneration are similar. These include oxidative stress, mitochondrial dysfunction, excitotoxicity, inflammatory changes, iron accumulation and protein aggregation [32,33,34].

Caffeine has neuroprotector and cytoprotector effects in cases of dopamine-induced damage by regulating the levels of this neurotransmitter [29]. Caffeine has also been associated with an effect that retards signs of neuronal degeneration in diseases like Parkinson’s and Alzheimer’s, effects that occur through interaction with adenosine receptors and a possible association with *N*-methyl d-aspartate receptors (NMDA), which increase cognition and delay the onset of the associated signs [35,36,37].

In addition, GSH depletion, disorders of cellular metabolism and cell death are known to be major mechanisms associated with oxidative and free radical damage in neuronal tissue. Three models of induced oxidative stress were applied—treatment with 6-hydroxydopamine (6-OHDA), *tert*-butylhydroperoxide (*t*-BuOOH) and iron-ascorbate (Fe^2+^/AA)—against which the possible protective effects of the studied hydrazide-hydrazone derivatives were evaluated.

The model of 6-OHDA-induced oxidative stress on isolated synaptosomes is based on the metabolism of 6-OHDA, leading to the production of reactive quinones (*p*-quinone), which in turn lead to the formation of reactive oxygen species (ROS). Reactive metabolites and ROS cause pre- and post-synaptic membrane damage and lead to neuronal cell damage [38].

To assess the impact of the studied substances, their effect on the parameters characterizing the functional-metabolic profile of synaptosomes—synaptosomal vitality—and reduced glutathione (GSH) levels was monitored, where the representatives showing the weakest neurotoxicity were screened.

A number of studies have shown that caffeine at a dose of 5 mg/kg increases plasma levels of GSH and total antioxidant potential, on the one hand, and on the other hand, reduces the levels of hydroperoxides and malonaldehyde [39]. Evidence suggests that improved GSH levels can be explained by increased cysteine production in the presence of caffeine [40].

The analysis of the results points out that the introduction of the hydrazone group is not critical for the protective properties of the studied structures. A more important factor appears to be the replacement of the tertiary carbon atom in the carbonyl part of the hydrazone molecule with a secondary one as in the structure of the target **6**, **6a**–**p**. The obtained results in this series led to the observation that the number and type of substituents in the phenyl fragment of the carbonyl part of the structure are important for the neuroprotective properties, since the presence of a large number of oxygen atoms has a beneficial effect.

Unsaturated fatty acids in the cell membrane are particularly sensitive to the action of ROS. Therefore, lipid hydroperoxides alter membrane density and membrane protein function. Lipid hydroperoxides, in turn, can undergo iron-mediated, one-electron reduction and oxidation to the formation of epoxyalyl peroxide radicals to trigger lipid peroxidation processes. The end products of lipid peroxidation are reactive aldehydes (e.g., 4-hydroxynonenal and malondialdehyde), which are toxic to the cell. A widely used experimental model for oxidative stress is the Fe^2+^/AA-induced lipid peroxidation model. The formation of reactive free radicals alters membrane integrity by inducing lipid peroxidation [41,42].

Caffeine has been shown to protect iron trichloride-treated mice in vivo from iron-induced oxidative damage. The authors believe that the antioxidant action of caffeine is due to increased levels of GSH on the one hand, and on the other hand its ability to capture ROS, in particular on ^•^OH [43]. However, caffeine can affect GSH metabolism [40].

The in vitro antioxidant properties of the tested series were studied in a model of lipid peroxidation induced by Fe^2+^/ascorbic acid (Fe^2+^/AA) on isolated liver microsomes from rats. This model system is commonly used to induce non-enzymatic lipid peroxidation. The effect was determined by measuring the content of malondialdehyde (MDA). Thus, it is assumed that the test compounds do not exhibit pro-oxidant activity. In contrast, treatment with Fe^2+^/AA causes a significant (threefold) increase in the amount of MDA compared to untreated controls.

The results from the non-enzyme-induced lipid peroxidation model identified compounds **6k** and **6l** as the best neuroprotective representatives, where the more pronounced effect was determined for **6l**, containing a free OH group, compared to **6k**, where the same position was substituted by a –OCH_3_ group.

The cytotoxic compound *tert*-butyl hydroperoxide (*t*-BuOOH) is frequently used as an amphiphilic lipid peroxidation-accelerating pro-oxidant in processes of the investigation of lipid peroxidation mechanisms. The *t*-BuOOH-induced toxicity is a chain reaction process attributed to the generation of butoxyl radicals via a Fenton-type reaction, causing oxidative damage and lipid peroxidation both in polyunsaturated and, at a lower extent, in monounsaturated fatty acids, mainly in biological membranes [44].

The cell was found to have defense mechanisms that reduce the toxic effects of *t*-BuOOH. Under conditions of oxidative stress from the endoplasmic reticulum are released so-called stress proteins Grp78, Grp94 and calreticulin. They block the increase in intracellular calcium, but do not block the lipid peroxidation process caused by ROS overproduction [45,46].

Literary data indicate that Caffeine significantly reduces the formation of endoplasmic reticulum stress proteins Grp78, Grp94 and calreticulin, which suggests that caffeine can protect cells against oxidative stress-induced changes to a wide spectrum of markers for ER stress [47]. These data confirmed the membrane-protecting abilities of caffeine from oxidative stress by its antioxidant effects [48,49,50] and its properties to inhibit lipid peroxidation induced by ROS [48].

These data determined our interest in evaluating the effect on the neuroprotective properties of caffeine after addition of a hydrazide-hydrazone group on the 8th position in the caffeinylthio-hydrazide molecule The results from the performed evaluations in the model of *t*-BuOOH-induced oxidative stress identified derivatives **6k** and **6l** to show a statistically significant neuroprotective effect on the toxic agent.

The performed analyses of the neuroprotective activity of the least toxic representatives of the newly synthesized xanthinylthio-hydrazide hydrazone derivatives on the three models of induced neurotoxicity (6-OHDA-induced oxidative stress, Fe^2+^/AA non-enzyme-induced lipid peroxidation and *t*-BuOOH-induced oxidative stress) performed at subcellular level (synaptosomes, microsomes and mitochondria) pointed out that the tested compounds showed a statistically significant neuroprotective effect. It is noteworthy that in all three evaluated models, compound **6l** was underlined with the highest neuroprotective effects. In addition, it was observed that the structure of the compounds exhibiting the strongest neuroprotective properties contained -OCH_3_ and/or -OH groups on *o*, *m* and *p*-positions in phenyl radicals, which we believe is due to the presence in the structure of a high number of strongly electronegative oxygen atoms.

### 3.4. Investigation of the Influence of the Newly Obtained Derivatives on the Activity of Human Recombinant MAO-B Enzyme

The analysis of the literature data shows the potential role of caffeine and xanthine derivatives as a possible adjuvant therapy in PD.

MAO-B amino oxidase specifically catalyzes the deamination of the false neurotransmitters benzylamine and β-phenylethylamine and is irreversibly inhibited by low concentrations of (R) -deprenyl [51]. The MAO-B enzyme is involved in the metabolism of dopamine in the human basal ganglia, which is why MAO-B inhibitors have been used successfully in the treatment of neurodegenerative diseases.

MAO-B inhibitors may exhibit neuroprotective effects by stoichiometric reduction in reaction products—aldehyde and hydrogen peroxide catalyzed by the enzyme MAO-B in the brain [52]. Therefore, the synthesis and creation of new biologically active substances with MAOB inhibitory activity is a promising direction in the prevention and treatment of neuronal pathologies.

Caffeine is a weak MAO inhibitor, but the synthesis of C-8 analogues is an essential step toward the preparation of potent and selective monoamine oxidase inhibitors [53]. The introduction of an electron withdrawing group at the styryl ring in the structure of (E)-8-styrylcaffeine pointed to an enhancement of the expected MAO-B inhibition, determined by the possible binding mode to MAOB enzyme by traversing the entrance and substrate cavities like a large-molecule inhibitor. In the active side of the enzyme, the polar functional groups of the caffeine ring are closed to flavine in the substrate cavity, and the styryl group extends into the entrance cavity. This dual binding is probably the reason for increasing MAO-B inhibition potency by caffeine’s derivatives [53]. In addition, a derived structure–activity relationship showed that inhibition potency increases with lipophilicity and bulkiness with benzyloxycaffeine derivatives performing highly selective MAO-B inhibition potencies. In addition, the published data suggested that the substitution of the oxygen atom into the sulfur atom in benzyloxycaffeine’s derivatives improves MAO-B inhibition potency [53].

These results gave us reason to investigate the possible inhibitory effects of the target newly synthesized xanthine derivatives on human recombinant MAO-B enzyme.

From studies on the inhibitory effect on human recombinant MAO-B enzyme (*h*MAOB) of the tested compounds, we found that compound **6l** exhibited inhibitory activity similar to that of Selegiline and more pronounced than that of caffeine. We believe that these findings are a result of the increased lipophilicity and introduction of the additional hydrazide-hydrazone group, together with the sulfur atom at the 8th position in the caffeine core.

## 4. Materials and Methods

### 4.1. Chemistry

All evaluated hydrazide-hydrazones were synthesized in the Faculty of Pharmacy, Medical University–Sofia, according to the previously described procedure [23] through condensation of equimolar amounts of 3-[(caffeine-8-yl)thio]propanohydrazide and differently substituted aromatic aldehydes. The reactions were carried out in dry alcoholic media. The solvent was removed under reduced pressure and the crude products were recrystallized from ethanol/water mixture (1:1). All obtained compounds were characterized by instrumental analytical procedures (UV-Vis, FTIR, NMR, MS) [23].

### 4.2. Biological Evaluation

#### 4.2.1. Animals

The experiments were performed on 6 male Wistar rats (body weight 200–250 g), which were housed in 2 organic glass cages with a 12/12 h light/dark cycle and standard laboratory conditions (ambient temperature 20 °C ± 2 °C and humidity 72% ± 4%) with free access to water and standard pelleted rat food 53-3, produced according to ISO 9001:2008. The experimental animals were acquired from the National Breeding Center, Sofia, Bulgaria. Seven days acclimatization was allowed before the commencement of the study and a veterinary physician monitored the health of the animals regularly. The vivarium (certificate of registration of farm No. 0072/01.08.2007) was inspected by the Bulgarian Drug Agency in order to check the husbandry conditions (No. A-11-1081/03.11.2011). All procedures were carried out in strict compliance with the requirements of the Institutional Committee for Animal Welfare and the principles set out in the European Convention for the Protection of Vertebrate Animals Used for Experimental and Other Scientific Purposes (ETS 123) (Council of Europe, 1991), throughout the experiment. The experiments with the animals were approved by the Bulgarian Agency of Food Safety with approval code 2200-0446, approval date 13 November 2017.

#### 4.2.2. Isolation and Incubation of Rat Brain Synaptosomes and Mitochondria

The synaptosomes and mitochondria were prepared as explained in [54,55] with the help of a colloidal silicon solution (Percoll): 1. Preparation of 90% stock solution of Percoll. 2. Preparation of Percoll solutions with two concentrations, 16% and 10%. An amount of 4 ml of Percoll 16% and 10% were added in the tubes. 3. An amount of 90% Percoll (7.5% Percoll) was added to the sediment from the last centrifugation. The tubes were centrifuged for 20 min at 15,000× *g* at 4 °C, resulting in the formation of three layers. The lower layer contained mitochondria, the upper layer, lipids and the middle layer (between 16% and 10% Percoll), synaptosomes. The middle and the lower layers of each tube were harvested and buffer B with glucose was added. The mixture was centrifuged at 10,000× *g* for 20 min at 4 °C. After centrifugation, the sediment with the synaptosomes and mitochondria was mixed with buffer B with glucose.

#### 4.2.3. Synaptosomal Viability

After the incubation, the synaptosomes were centrifuged three times at 15,000× *g* for 1 min. An MTT test was performed to determine synaptosomal vitality by the method described in the literature [56].

#### 4.2.4. Determination of Reduced Glutathione (GSH) in Brain Synaptosomes

The level of reduced glutathione was determined by measuring the non-protein SH-groups after precipitation of the proteins with trichloroacetic acid. After the incubation, synaptosomes were centrifuged at 400× *g* for 3 min. These diment were treated with 5% trichloroacetic acid and left for 10 min on ice. Samples were centrifuged at 8000× *g* for 10 min (2 °C). The supernatant was removed to determine the level of GSH and was stored at −20 °C. Immediately before the measurement, the samples were neutralized with 5 N NaOH.

The presence of thiols in the supernatant was determined using Ellman’s reagent. The resulting yellow color was measured spectrophotometrically at λ = 412 nm [57].

#### 4.2.5. Model of 6-OHDA-Induced Neurotoxicity in Synaptosomes

This in vitro model resembles the neurodegenerative processes occurring in PD. Dopamine metabolism and oxidation lead to the formation of ROS and reactive quinones. They induce dopamine neurotoxicity and neurodegeneration [38]. The synaptosomes were incubated with 150 μmol 6-OHDA for 1 h [58].

#### 4.2.6. Tert-Butyl Hydroperoxide (t-BuOOH)-Induced Oxidative Stress in Isolated Brain Mitochondria

Amounts of 0.3 ml 0.2% TBA and 0.25 ml sulfuric acid (0.05 M) for incubation of mitochondrial suspensions (0.5 mg protein/mL) were applied for 30 min. After that, the tubes were maintained in a bath filled with boiling water. Finally, the tubes were transferred to an ice bath and butanol (0.4 mL) was poured into each tube. In the next stage, centrifugation of the tubes was carried out at 3500× *g* for 10 min. Following that, assessment of the total quantity of MDA shaped in each sample was carried out using a spectrophotometer (UV-Vis Split) to measure the supernatant’s absorbance at 532 nm. The mitochondria were incubated with 75 µmol *t*-BuOOH [59].

#### 4.2.7. Measurement of GSH Content in Brain Mitochondria

The DNTB spectrophotometric technique explained in [60] was applied for GSH content measurement in brain mitochondria. Then, 0.04% DTNB was mixed with the mitochondrial suspensions (0.5 mg protein/mL) in 0.1 mol/L of phosphate buffers (pH 7.4). The development of the yellow color was read at 412 nm (spectrophotometer UV-Vis Split).

#### 4.2.8. Preparation of Brain Microsomes Using Ultracentrifugation 

The brain microsomes were prepared according to the procedure described in [61]. The brain was homogenized in 9 vol of 0.1 M Tris containing: 0.1 mM Dithiothreitol, 0.1 mM Phenylmethylsulfonyl fluoride, 0.2 mM EDTA, 1.15% potassium chloride and 20% (*v/v*) glycerol at pH 7.4. The homogenate was centrifugated at 17,000× *g* for 30 min. The pellet was suspended in 4 vol of the homogenization buffer and centrifuged at 17,000× *g* for a further period of 30 min. The supernatants from both centrifugations were combined (post-mitochondrial supernatant) and centrifuged at 100,000× *g* for 1 h. The pellet was resuspended in the above buffer and centrifuged at 100,000× *g* for 1 h. The resulting pellet was frozen in the buffer with glycerol.

#### 4.2.9. FeSO_4_/Ascorbic Acid-Induced Lipid Peroxidation in Isolated Brain Microsomes

The microsomes were pre-incubated with the analyzed compounds for 15 min at 37 °C. The reaction was started with a solution of iron sulphate 20 μmol and ascorbic acid 0.5 mmol. The reaction was stopped with a mixture of TCA 25% and TBA 0.67% at 20 min. After starting the reaction, the LPO and the MDA quantity was assessed [62].

#### 4.2.10. MDA Assay in Brain Microsomes and Mitochondria

The quantity of the lipid peroxidation product MDA was assessed, using the method, described in the literature [62]. The absorbance was measured at 535 nm, and the amount of MDA was calculated using a molar extinction coefficient of 1.56 × 10^5^ mol^−1^ cm^−1^.

#### 4.2.11. Measurement of Monoamine Oxidase B Activity

Monoamine oxidase activity assay of recombinant human MAO-B was performed using a fluorimetric method by Amplex Ultra Redreagent [63]. Tyramine hydrochloride was used as substrate.

### 4.3. Quantitative Structure–Neurotoxicity Assessment for the Evaluated Xanthinylthio Propanehydrazones

Analysis of quantitative structure–neurotoxicity relationships employed the data from in vitro assessments of synaptosome viability (%) and MDA production in microsomal fractions (nmol/0.1 mg protein) as indicators for the neurotoxicity of compounds. Effects of the newly synthesized derivatives were compared to the effect of caffeine (**1**) and expressed in terms of relative synaptosomal viability (RV) and relative increase in MDA production (RMDA), using the equation:(3)$$Y_{test}=\frac{Y_{in\;vitro}}{Y_{positive\;control}}$$
where *Y**_test_* stands for calculated values of RV and RMDA; *Y**_in vitro_* are in vitro measured values of synaptosomal viability (%) and MDA production; *Y**_positive control_* accounts for the same effects measured after treating of synaptosomes and microsomes with caffeine 1 as a positive control.

The structures of xanthinylthio propanehydrazones were built and optimized by MM+ force field [64], implemented in HyperChem 7.5 (HyperCube Ltd., London, UK). The same software was used for calculation of the electron density distribution by AM1 method [65] and a set of electronic descriptors. LogP values were calculated by ACD Labs software (ACD Inc., Walnut, CA, USA). A number of descriptors concerning 2D and 3D properties of molecules were generated by MDL QSAR software, version 2.2 (Symyx). The variable selection was performed by a genetic algorithm (GA) [66] set at an initial population size of 32, uniform crossover, one-point mutation and Friedman’s lack-of-fit scoring function. Multiple linear regression (MLR) was applied for generation of QSTR equations. The models were estimated on the basis of explained variance *r*^2^, standard error of estimate SEE, F-ratio, cross-validated correlation coefficient *q*^2^ from a leave-one-out procedure and *r*^2^*_sc_* after a randomization test (Y scrambling test).

### 4.4. Statistical Analysis

The MAO-B activity was normalized as a percentage of the untreated control set as 100% and the results were expressed as mean values and standard deviation (±SD). Statistical analysis was performed by one-way analysis of variance (ANOVA) with a post hoc multiple comparisons procedure (Dunnet’s test) to assess the statistical differences in case of normal distribution. Values of *p* < 0.05, *p* < 0.01 and *p* < 0.001 were considered as statistically significant.

Statistical analysis of the results, obtained from brain microsomes, synaptosomes and mitochondria, were performed using the statistical program “MEDCALC”. Results are expressed as mean ± SEM for 6 experiments. The significance of the data was assessed using the non-parametric Mann–Whitney test. Values of *p* ≤ 0.05, *p* ≤ 0.01 and *p* ≤ 0.001 were considered statistically significant.

## 5. Conclusions

The obtained results regarding the studied neurotoxicity assessed at a subcellular level on isolated cell synaptosomes, microsomes and mitochondria, determined by measuring the influence of the newly designed series of *N*′-substituted 3-(1,3,7-trimethyl-xanthin-8-ylthio)propanehydrazides on cell viability and GSH levels and MDA production, identified compounds **6k** and **6l** with the least pronounced neurotoxic effects. In the evaluation of the neuroprotective effects of the least toxic newly synthesized methylxanthine derivatives, performed in three models of induced oxidative stress—6-OHDA- and *t*-BuOOH-induced oxidative stress and Fe^2+^/AA non-enzyme-induced lipid peroxidation, compounds **6k** and **6l** showed the best protective effects. From experiments performed on human recombinant MAO-B enzyme (*h*MAOB), we found that substance **6l** exhibited inhibitory activity similar to that of Selegiline and more pronounced than that of caffeine. The same substance also showed a pronounced neuroprotective effect in all three models of induced oxidative stress, which we assume was due not only to the preservation of GSH levels, but also to the inhibition of MAO-B activity.

The conducted biological tests highlight, as the most promising compounds for future in vivo research and leaders in the design of new xanthine derivatives affecting neurodegenerative pathologies, 2 out of a total of 17 not described in the literature, methylxanthine derivatives **6k** and **6l**.

The performed QSTR study revealed that neurotoxicity of the investigated xanthinylthio propanehydrazones is mainly dependent on their lipophilicity and polarity. Since both the properties magnify this effect, a potentially useful improvement strategy is to introduce hydrophilic and less polar substituents in the structure. Thus, the eventual further investigations will focus on favorable structural modifications with an emphasis on the number, type and mutual arrangement of substituents.

## Figures and Tables

**Figure 1 molecules-27-05321-f001:**
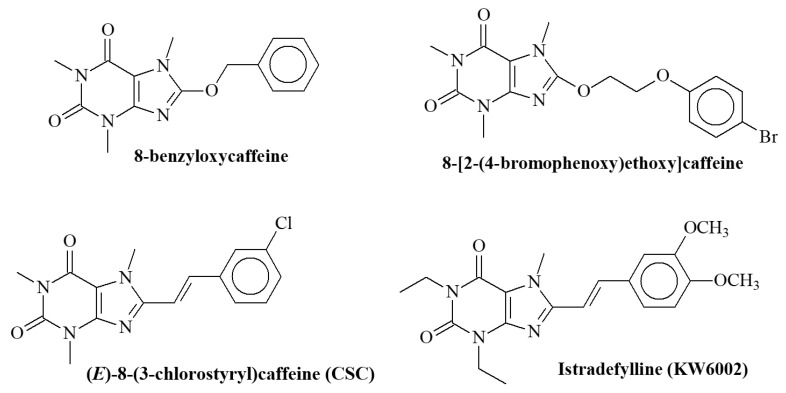
The structures of some xanthine derivatives with potent reversible inhibitor activity on human monoamine oxidase (*h*MAOB).

**Figure 2 molecules-27-05321-f002:**
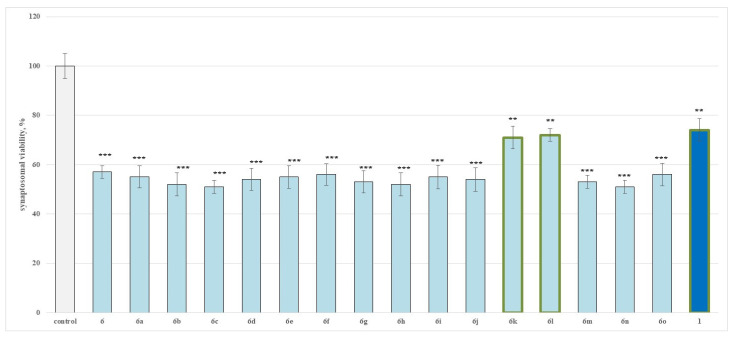
Effects of compounds **6**, **6a**–**o** and **1** (100 µmol), applied alone, on synaptosomal viability. ** *p* < 0.01; *** *p* < 0.001 against the control (untreated synaptosomes). The green outline indicates the most promising derivatives.

**Figure 3 molecules-27-05321-f003:**
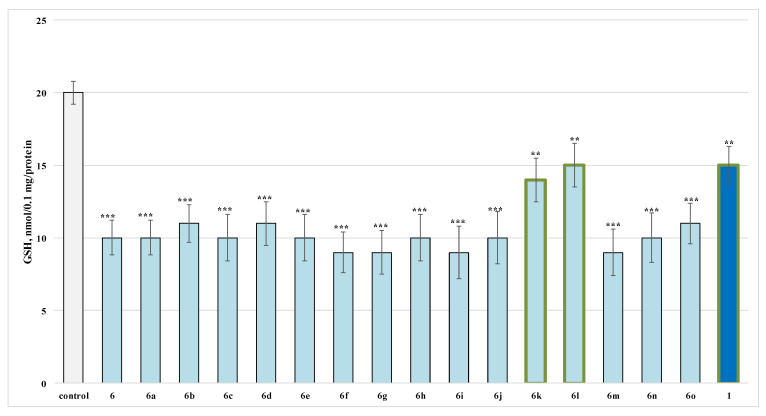
Effects of compounds **6**, **6a**–**o** and **1** (100 µmol), applied alone, on GSH levels. ** *p* < 0.01; *** *p* < 0.001 against the control (untreated synaptosomes). The green outline indicates the most promising derivatives.

**Figure 4 molecules-27-05321-f004:**
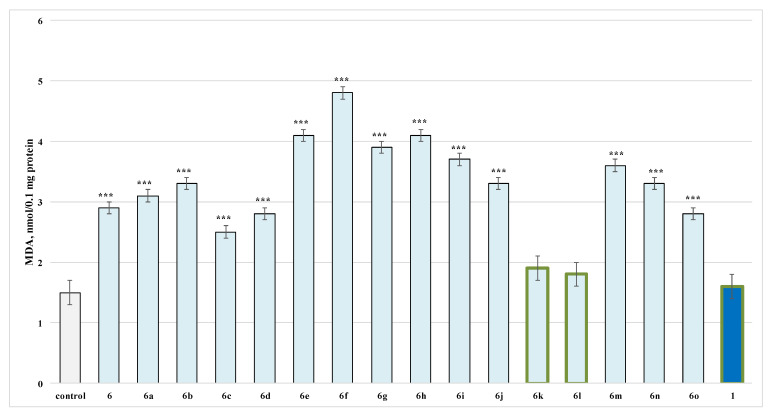
Effects of compounds **6**, **6a**–**o** and **1** (100 µmol), applied alone, on malonaldehyde (MDA) production in isolated rat brain microsomes. *** *p* < 0.001 against the control (non-treated microsomes). The green outline indicates the most promising derivatives.

**Figure 5 molecules-27-05321-f005:**
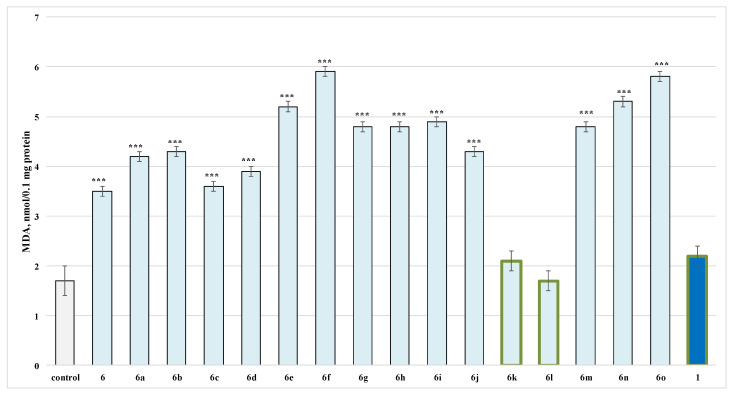
Effects of compounds **6**, **6a**–**o** and 1 (100 µmol), applied alone, on MDA production in isolated rat brain mitochondria. *** *p* < 0.001 against the control (untreated mitochondria). The green outline indicates the most promising derivatives.

**Figure 6 molecules-27-05321-f006:**
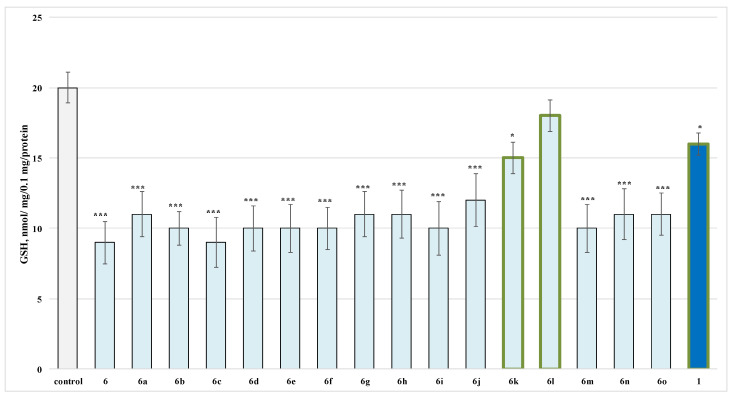
Effects of compounds **6**, **6a**–**o** and **1** (100 µmol), applied alone, on GSH level in isolated rat brain mitochondria. * *p* < 0.05, *** *p* < 0.001 against the control (untreated mitochondria). The green outline indicates the most promising derivatives.

**Figure 7 molecules-27-05321-f007:**
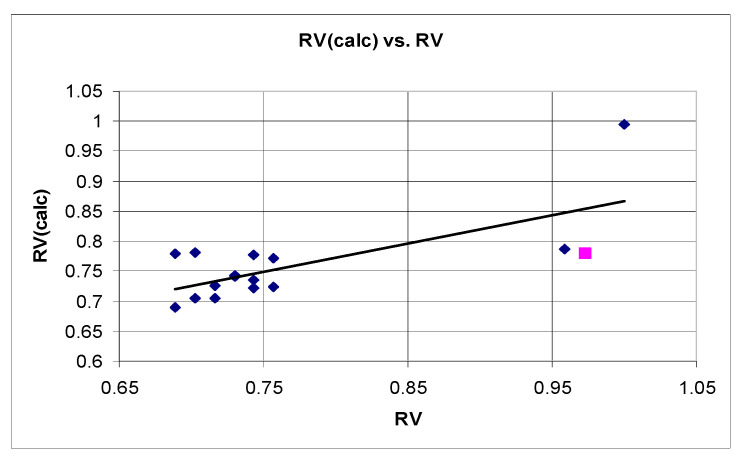
Correlation between the experimentally determined and calculated values of the relative synaptosomal vitality (RV) for the investigated series. The “outlier” **6l** is given in pink.

**Figure 8 molecules-27-05321-f008:**
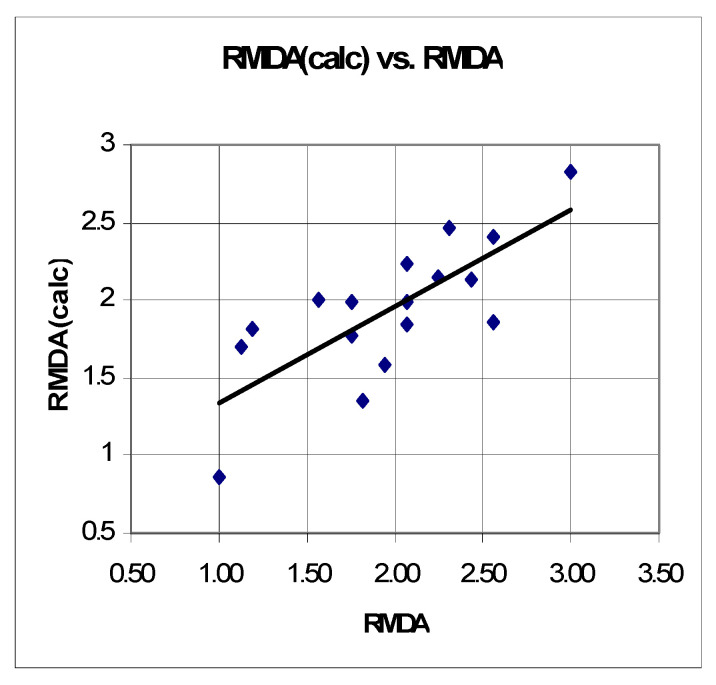
Correlation between the experimentally determined and calculated values of the relative increase in MDA production (RMDA) for the investigated series.

**Figure 9 molecules-27-05321-f009:**
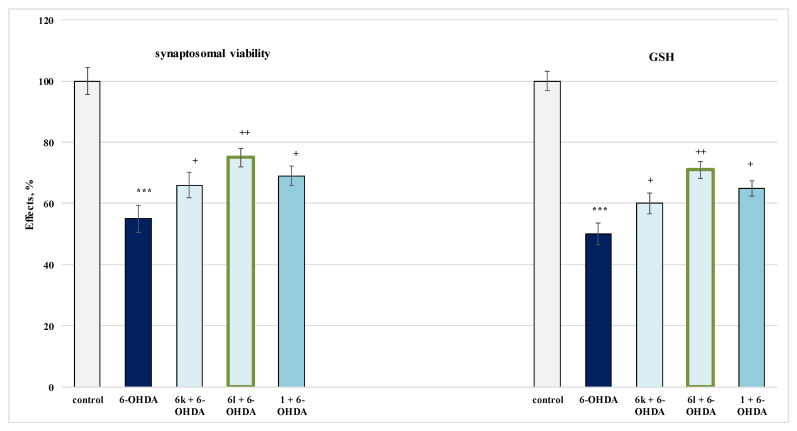
Effects of **6k**, **6l** and **1** in a model of 6-OHDA-induced oxidative stress on synaptosomal vitality. *** *p* < 0.001 against the control (untreated synaptosomes); ^+^
*p* < 0.05; ^++^
*p* < 0.01 against 6-OHDA. The green outline indicates the most promising derivatives.

**Figure 10 molecules-27-05321-f010:**
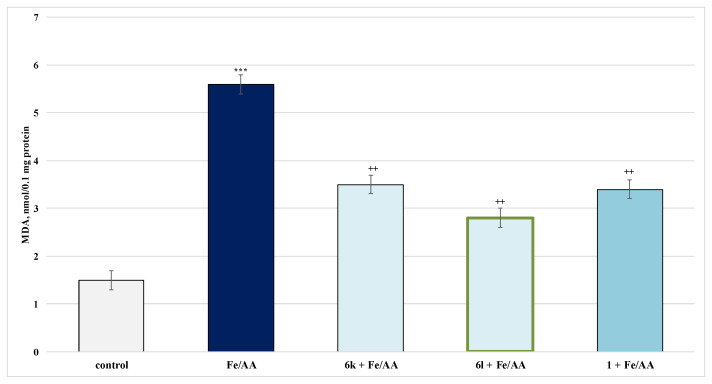
Effects of the tested **6k**, **6l** and **1** (100 µmol) in a model of enzyme-induced lipid peroxidation on the production of MDA in isolated rat microsomes. *** *p* < 0.001 against the control (untreated microsomes); ^++^
*p* < 0.01 against Fe^2+^/AA. The green outline indicates the most promising derivatives.

**Figure 11 molecules-27-05321-f011:**
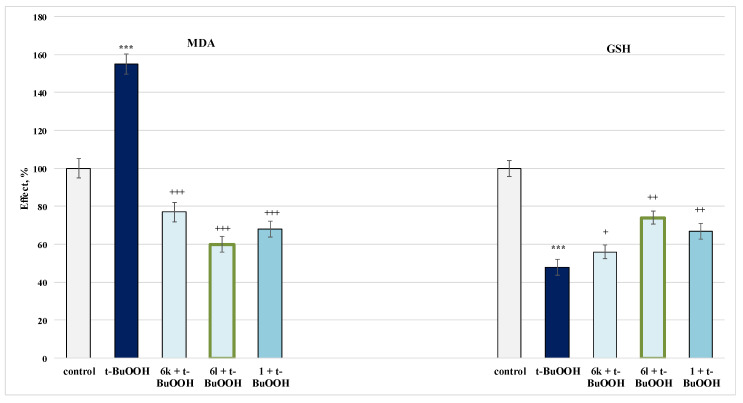
Effects of the tested **6k**, **6l** and **1** (100 µmol) in a model of *t*-BuOOH-induced oxidative stress on the MDA production and GSH level in isolated rat mitochondria. *** *p* < 0.001 against the control (untreated mitochondria); ^+^
*p* < 0.05; ^++^
*p* < 0.01; ^+++^
*p* < 0.001 against *t*-BuOOH. The green outline indicates the most promising derivatives.

**Figure 12 molecules-27-05321-f012:**
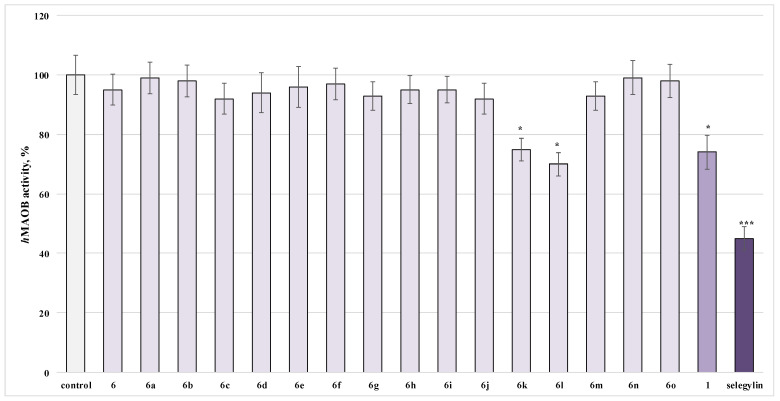
Effects of the target **6**, **6a**–**o**, **1** and Selegiline (1 µmol) applied alone, on the *h*MAOB activity. * *p* < 0.05; *** *p* < 0.001 against the control (pure *h*MAOB).

**Table 1 molecules-27-05321-t001:** ID and structures of the studied compounds.

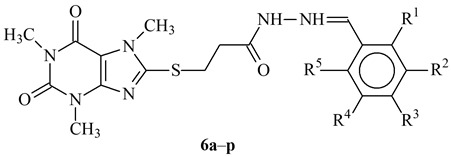						
Compound	R^1^	R^2^	R^3^	R^4^	R^5^	R^6^
**6a**	H	H	H	H	H	H
**6b**	H	H	NO_2_	H	H	H
**6c**	Cl	H	H	H	Cl	H
**6d**	H	H	H	Cl	H	H
**6e**	H	H	Br	H	H	H
**6f**	H	H	CF_3_	H	H	H
**6g**	H	OCH_3_	OCH_3_	H	H	H
**6h**	H	OCH_3_	OCH_3_	OCH_3_	H	H
**6i**	H	OCH_3_	OH	OCH_3_	H	H
**6j**	H	H	OH	OCH_3_	H	H
**6k**	OCH_3_	OCH_3_	H	H	H	H
**6l**	OH	H	H	H	H	H
**6m**	CH_3_	H	CH_3_	H	CH_3_	H
**6n**	OH	OCH_3_	H	H	H	H
**6o**	H	NO_2_	H	H	H	H
**6p**	H	NO_2_	OH	OCH_3_	H	H

**Table 2 molecules-27-05321-t002:** Calculated values for the relative synaptosomal vitality (RV), relative increase in the MDA production (RMDA) and significant descriptors for the evaluated neurotoxicity.

Compound ID	RV	RMDA	LogP_ACD	Dipole (D)	xc3
**1**	1.000	1.000	−0.13	2.657	1.253
**6a**	0.743	1.938	3.410	8.639	1.869
**6b**	0.703	2.063	3.330	8.706	2.369
**6c**	0.689	1.563	4.800	11.438	2.272
**6d**	0.730	1.750	4.210	7.708	2.158
**6e**	0.743	2.563	4.340	9.912	2.158
**6f**	0.757	3.000	4.380	9.161	3.379
**6g**	0.716	2.438	3.530	16.350	2.202
**6h**	0.703	2.563	3.400	20.764	2.338
**6i**	0.743	2.313	2.790	21.067	2.39
**6j**	0.730	2.063	3.040	17.462	2.271
**6k**	0.959	1.188	3.130	9.319	2.134
**6l**	0.973	1.125	3.490	7.687	2.067
**6m**	0.716	2.250	4.790	8.943	2.561
**6n**	0.689	2.063	3.360	8.788	2.191
**6o**	0.757	1.750	3.510	8.591	2.369

## Data Availability

Not applicable.

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
