# Peer review of "Quantitative Structure–Neurotoxicity Assessment and In Vitro Evaluation of Neuroprotective and MAO-B Inhibitory Activities of Series N′-substituted 3-(1,3,7-trimethyl-xanthin-8-ylthio)propanehydrazides"

_molecules, 2022, doi:10.3390/molecules27165321_

Round 1

Reviewer 1 Report

This interesting work describes the neurotoxicity, neuroprotective and MAOB inhibitory effects of series N’-substituted 16 3-(1,3,7-trimethyl-xanthin-8-ylthio)propanehydrazide.

Substances exhibiting these properties can be good drug candidates for the treatment of neurological diseases such Parkinson's Disease. Unfortunately, the work requires re-editing

First of all, the discussion is poorly written. This chapter mainly describes the methods used (this can optionally be included in the Materials and Methods). The discussion should consist in comparing the obtained results with the current state of knowledge on the subject. References to current literature are missing in this chapter. There are only citations regarding the methods described. I don't understand why chapter 3.3 is under discussion, it should be moved to the Materials and Methods and partly to the Results. All the more so as the results of these methods are presented, and they are not described in the Materials and Methods. Moreover, the Methods used are described very briefly. Moreover, the Methods used are described very briefly. In this chapter, there is no description of the methods which, as I mentioned earlier, are described in the Results and Discussion. Coming back to the discussion, may it be associated with the results? Caffeine results once full name is used once 1, no explanation in figure description. In the description of the figures, caffein is given as 1 or its full name. Please standardize it.

In conclusion, despite the interesting topic and the use of many modern and interesting methods, the work in this form is not suitable for publication in the Molecules journal.

Reviewer 2 Report

This work presents a series of biological tests along with a QSTR study in order to investigate the neurotoxicity, neuroprotective and MAOB inhibitory effects of a series of N’-substituted 3-(1,3,7-trimethyl-xanthin-8-ylthio)propanehydrazides.

Unfortunately, the QSAR model based on the calculated relative synaptosomal vitality (RV) data have some shortcomings:

-          the structure 6l (the best experimentally identified compound) was computationally identified as „outlier”, and thus its neurotoxicity cannot be predicted with sufficient accuracy.

-          the values of r2 and q2 for the linear equations 2 and 3 are quite weak, especially for equation 3. According to Alexander Golbraikh and Alexander Tropsha, a QSAR model is predictive, if the following conditions are satisfied: q2 > 0.5; R2 > 0.6; [Golbraikh, A. and Tropsha, A., J. Mol. Graphics Model. 20 (2002) 269–276.] [Golbraikh, A. and Tropsha, A., Journal of Computer-Aided Molecular Design, 16: 357–369, 2002.]

-          lines 339-340: “Respectively, the least toxic 6k and 6l, together with Caffeine 1, have the lowest values for the parameter Dipole.” This statement is not exactly correct according to Table 2. There are many other compounds in the table 2 (6a, 6b, 6d, 6f, 6m, 6n, 6o) with dipole values lower than 6k.

Nevertheless, it can be said that the study is interesting and can represent a starting point for more investigations on this subject.

Some suggestions:

It would be indicated to add in the Introduction a paragraph that expresses the purpose of the study and the approaches used.

Please put the explanation of the abbreviations where it appears for the first time in the text, e.g.

-          MDA (line 124), and delete “malondialdehyde” from line 147.

-          ROS (line 390), and delete “reactive oxygen species” from line 521.

Correct the typo “caffein e” (line 49).

Round 2

Reviewer 1 Report

Thanks to the authors for including my suggestions in the manuscript. I have one more suggestion for the Discussion chapter. Could the authors combine this chapter as a whole, without subchapters, or with the most important ones, such as: 3.2 to 3.5.? Without further subdivision? If I would delete this section 3.1, this information is in the Materials and Methods section.

Author Response

We think that in point of clarity would be good if the Discussion part is divided to subchapters. However we agree that the further subdivision is unnecessary, so we deleted subsection 3.1. and combined all the other subdivisions, leaving only the important subchapters 3.2 to 3.5 as suggested by the Reviewer.